# A Combination Native Outer Membrane Vesicle (NOVM) Vaccine to Prevent Meningococcal and Gonococcal Disease

**DOI:** 10.3390/pathogens14100979

**Published:** 2025-09-26

**Authors:** Serena Giuntini, Scarlet W. Tefera, Alejandro Bolanos, Adan Ramos Rivera, Gregory R. Moe

**Affiliations:** OMVax, Inc., San Francisco, CA 94111, USA; scarlet@synergenics.net (S.W.T.); alejandro@synergenics.net (A.B.); adan@synergenics.net (A.R.R.); greg@synergenics.net (G.R.M.)

**Keywords:** factor H binding protein, porin outer membrane protein B, carcinoembryonic antigen-related cell adhesion molecule 1, colonization, transgenic mouse model

## Abstract

The increase in the incidence and antibiotic-resistant strains show a need for a broadly protective vaccine to prevent gonorrhea. OMVax has developed a combination vaccine based on native outer membrane vesicles (NOMVs) from two *Neisseria meningitidis* (Nm) and two *Neisseria gonorrhoeae* (Ng) strains. The strains had the acyl transferase LpxL1 knocked out to increase safety, and the reduction-modifiable protein was also knocked out in the Ng strains. Factor H binding protein (FHbp) mutants with reduced Factor H (FH) binding from Subfamilies A and B, respectively, were overexpressed in the Nm strains. The Ng strains individually expressed porin outer membrane protein B 1a (PorB.1a) or PorB.1b. Antibodies elicited by the Nm-Ng NOMV vaccine had SBA with a human complement against diverse Nm and Ng strains grown in the presence of Cytidine-5′-monophospho-N-acetylneuraminic acid (CMP-NANA), had no significant reduction in serum bactericidal activity (SBA) compared to the respective individual vaccines, inhibited the adhesion to human cervical and vaginal cells in five out of six Ng strains tested, and inhibited Nm and Ng colonization in a transgenic mouse model. In conclusion, the Nm-Ng NOMV vaccine has the potential to protect against disease and inhibit colonization by diverse Nm and Ng strains, which may be an advantage for controlling the disease through vaccination, particularly in the adolescent/young adult age group.

## 1. Introduction

Gonorrhea, caused by the bacterial pathogen *Neisseria gonorrhoeae* (Ng), is currently the second most common sexually transmitted infection (STI) world-wide with an incidence of >600,000 cases per year in the US and ~82 million globally. Ng colonizes the mucosa of men and women but disproportionately affects the health of women, resulting in reduced fertility and other complications of the upper and lower genital tract. The general increase in the incidence of the disease and the emergence of multi-drug-resistant strains has increased concerns for the potential of untreatable gonorrhea. While there is no vaccine specifically designed to prevent gonorrhea, epidemiological studies conducted by Petousis-Harris and co-workers in New Zealand revealed that a detergent-extracted outer membrane vesicle vaccine (dOMV) produced from closely related *Neisseria meningitidis* bacteria (MeNZB) had a 31% efficacy in preventing gonorrhea [1]. The MeNZB vaccine was developed by Chiron, Corp. as a tailor-made vaccine to control a meningococcal disease epidemic caused by *Neisseria meningitidis* serogroup B (MenB) [2]. The effect of MeNZB on reducing gonorrhea in New Zealand was subsequently confirmed for other MenB OMV-containing vaccines [3,4,5], including the current market leading MenB vaccine, 4CMenB (Bexsero^®^) [6], and led to the development of several new vaccines for Ng based on Ng OMVs [7,8,9]. However, a recent Phase 1/2 clinical trial of an Ng OMV vaccine produced by Galaxo Smith Kline (GSK) failed to meet pre-defined efficacy criteria [10].

Currently there are two MenB vaccines licensed in the US, 4CMenB and MenB-FHbp (Trumenba^®^). Both MenB vaccines, however, provide incomplete meningococcal strain coverage and elicit protection for a relatively short duration [11,12]. In particular, 4CMenB was predicted to not provide protection against strains, causing outbreaks in countries where it is used for the routine vaccination of infants, such as in the UK, France, Spain, and Canada. Recently, the MenB vaccine strain coverage in these countries was reported to be as little as 5.3% against emerging clonal complexes based on predictions by the genetic Meningococcal Antigen Typing System [13].

Recognizing the limitations of current MenB vaccines, OMVax has developed a next-generation MenB vaccine based on native outer membrane vesicles (NOMVs) with over-expressed mutant Factor H binding proteins (FHbps) from Subfamilies A and B with reduced Factor H binding [14,15]. In addition, the NOMVs were made safer by eliminating sialic acid antigens that mimic host antigens and by deleting the acyl transferase LpxL1, resulting in penta-acyl lipooligosaccharide (LOS), which has reduced endotoxin activity [7]. This latter modification eliminates the need for detergent extraction, which can remove important lipoprotein antigens, such as FHbp from OMV, and alter the structure of membrane protein epitopes that elicit protective antibodies [7].

Recent studies have shown that antibodies elicited by 4CMenB that are reactive with diverse Ng strains are directed to the MenB porin protein, PorB, and lipooligosaccharide (LOS) [16]. Since the MenB OMV has been shown to have a 30–40% efficacy against Ng in fully vaccinated adults, OMVax considered whether the protection against Ng by the MenB OMV could be increased by also including an NOMV component produced from Ng strains that express porin proteins that are representative of each of the two subfamilies of gonococcal porins, PorB.1a and PorB.1b (Ng NOMV). We show here that a combination MenB NOMV vaccine component (Nm NOMV) with overexpressed mutant FHbp from Subfamilies A and B provides broader coverage and higher serum bactericidal activity (SBA) against MenB than existing MenB vaccines and, when combined with Ng NOMV components (Nm-Ng NOMV), elicits antibodies with broad SBA against MenB and Ng strains and inhibits the colonization of Ng in human cervical and vaginal cell lines and in a transgenic mouse model of gonorrhea. Further, a combined MenB-Ng vaccine that protects against both pathogens may have greater acceptance than a stand-alone Ng vaccine, since MenB vaccines are routinely administered to control meningococcal disease in schools, colleges, universities, and the military in adolescents and young adults who have a higher risk of acquiring new Ng infections compared to other age groups [17].

## 2. Materials and Methods

The MenB vaccine component comprised NOMV naturally blebbed from two mutant strains of *N. meningitidis*. Each strain was derived from an H44/76 parent strain. Each strain has deletions of the wild-type *fhbp*, *lpxL1*, and *siaD-galE* genes by insertion of DNA-containing mutant *fhbp* genes (Subfamily A containing three amino acid substitutions or Subfamily B containing a single amino acid substitution), along with an engineered promoter and drug resistance markers (erythromycin, kanamycin, and spectinomycin, respectively). The Subfamily A and Subfamily B mutants of Factor H binding protein (FHbp) have reduced human Factor H (FH) binding [14,15]. Deletion of *lpxL1* results in penta-acylated lipooligosaccharide (LOS), which has attenuated endotoxin activity, compared with the wildtype hexa-acylated LOS [18]. Interruption of the *siaD* and *galE* genes eliminates expression of the group B capsular polysaccharide and LOS derivatives, which may elicit antibodies cross-reactive with human tissues [19,20]. To enhance FHbp expression further, each strain also carries a modified version of a multi-copy plasmid containing a chloramphenicol resistance gene [21] and coding for the respective mutant FHbp driven by the engineered promoter. The gonococcal vaccine component comprised NOMV naturally blebbed from two mutant strains of *N. gonorrhoeae*. Both *N. gonorrhoeae* strains, FA190 and WHO F, had LpxL1 and reduction-modifiable protein knocked out.

NOMV were purified from culture supernatant of bacteria grown in chemically de-fined media, for *N. meningitidis*, or GCLB [22] for *N. gonorrhoeae* with supplements that enable the bacteria to grow to high density. After reaching stationary phase, bacteria were harvested by centrifugation. The culture supernatant was filter sterilized using 0.2 µm filters and concentrated using 100 KDa filters (ThermoFisher Scientific, Waltham, MA, USA). NOMVs were collected by ultracentrifugation and further purified by size exclusion chromatography (ToyoPearl HW-55F; Millipore-Sigma, Burlington, MA, USA). Total protein concentration was measured by BCA (Pierce). The combination vaccine contained a 1:1 mixture of NOMV from *N. meningitidis* and *N. gonorrhoeae*. The combination vaccine referred to as Nm-Ng NOMV was combined with aluminum hydroxide adjuvant (Alhydrogel^®^ 2%, from Croda, Athens, AL, USA).

Five- to six-week-old CD-1 mice (*N* = 8 per group) were immunized I.P. on days 0 and 21 with Nm NOMV at 5 µg/dose, Ng NOMV at 5 µg/dose, and Nm-Ng NOMV at 10 µg/dose, all adjuvanted with 50 µg/dose of Al(OH)_3_. Adjuvant alone was given at 50 µg/dose as negative control. Blood was collected by terminal cardiac puncture on day 35. Human CEACAM-1 and FH transgenic FVB-BALB/c hybrid mice were immunized I.P. on days 0, 21, and 42 with Ng NOMV at 5 µ/dose and Nm-Ng NOMV at 10 µg/dose—both adjuvanted with 50 µg/Dose of Al(OH)_3_. Bexsero^®^ was administered at 1/5 of the human dose. Blood was collected by terminal cardiac puncture on day 56.

Binding of the postimmunization mouse sera to wild-type rFHbps and purified NOMV was measured by enzyme-linked immunosorbent assay (ELISA). The wells of a microtiter plate (Immulon 2B; ThermoFisher Scientific, Waltham, MA, USA) were coated with 2 µg/mL of rFHbp or NOMV in PBS and incubated overnight at 4 °C. The plates were blocked with PBS containing 0.1% Tween-20 (Millipore-Sigma, Burlington, MA, USA) (PBST) and 1% (weight/volume) bovine serum albumin (BSA) (Millipore-Sigma, Burlington, MA, USA). Concentration-dependent binding of the anti-FHbp and anti-NOMV Abs was measured at 1:100 to 1:200,000 dilution in PBST-BSA. After incubation (2 h at room temperature), the plates were washed and rabbit anti-mouse immunoglobulin G (IgG)-alkaline phosphatase (Zymed; 1:5000 diluted in PBST/BSA) was added. After 1 h at room temperature, alkaline phosphatase substrate (Millipore-Sigma, Burlington, MA, USA) was added, and the absorbance at 405 nm was measured after 30 min. ELISA titers were defined as the serum dilution, producing an OD_405 nm_ of 1.

For Nm serum bactericidal activity (SBA) assay, bacteria were grown to early log phase in liquid Frantz medium supplemented with 4 mM d,l-lactate (Millipore-Sigma, Burlington, MA, USA) and 2 mM cytidine 5′-monophospho-*N*-acetylneuraminic acid (CMP-NANA; Biosynth, Woburn, MA, USA) to enhance the sialylation of lipooligosaccharide [23]. Test sera were heated for 30 min at 56 °C to inactivate endogenous complement. The 40 μL reaction mixture used to measure bactericidal activity contained 2-fold serial dilutions of the test serum sample, 300 to 400 CFUs of bacteria, and 30% IgG-IgM-depleted human complement (Pel-Freez Biologicals). The SBA titer was the interpolated dilution, resulting in 50% survival of the bacteria after 60 min incubation at 37 °C and 5% CO_2_ compared to the number of CFUs per milliliter in negative control sera and complement.

For Ng SBA assay, bacterial cultures were prepared the day prior to testing by streaking frozen stocks onto Chocolate II Agar (GC II Agar with Hemoglobin and IsoVitaleX™, ThermoFisher Scientific, Waltham, MA, USA) and re-passaged onto fresh Chocolate II Agar supplemented with CMP-NANA (50 µg/mL) on the day of assay. After 6 h at 37 °C in an atmosphere containing 5% CO_2_, bacteria were then suspended in HBSS containing 1 mM MgCl_2_ and 0.15 mM CaCl_2_ and 2% BSA. Test sera were heated for 30 min at 56 °C to inactivate endogenous complement. The 75 μL reaction mixture used to measure bactericidal activity contained 2-fold serial dilutions of the test serum sample, 800 to 1000 CFUs of bacteria, and 20% (volume/volume) IgG-IgM-depleted human complement (Pel-Freez Biologicals). The SBA titer was the interpolated dilution, resulting in 50% survival of the bacteria after 60 min incubation at 37 °C and 5% CO_2_ compared to the number of CFUs per milliliter in negative control sera and complement.

IgG from postimmunization serum pools (*N* = 4 per pool) from mice immunized with each vaccine and adjuvant control were purified using Protein G HP SpinTrap columns (Cytiva Lifesciences, Marlborough, MA, USA). The concentrations of the purified mouse serum IgG were determined by a capture ELISA in which a goat anti-mouse κ chain-specific antibody (MyBiosource, San Diego, CA, USA) was adsorbed to the wells of a microtiter plate and used to capture the Abs. The bound Abs were detected with goat anti-mouse IgG (H) antibody conjugated with alkaline phosphatase (Jackson ImmunoResearch, West Grove, PA, USA). The concentrations of the mouse IgG Abs were assigned by comparison with binding of a purified mouse IgG standard (BioRad, Hercules, CA, USA).

For colonization assays, bacterial cultures were prepared the day prior to testing by streaking frozen stocks onto Chocolate II Agar (GC II Agar with Hemoglobin and IsoVitaleX™, ThermoFisher Scientific, Waltham, MA, USA). Bacteria were then suspended in PBS to an OD_620 nm_ of 0.3, and 10 µM CellTrace™ CFSE Cell Proliferation dye (Invitrogen, Waltham, MA, USA) was added to the solution. The labeling reaction was stopped after 30 min with fetal bovine serum (FBS), and free label was removed with two washes. Labeled bacteria were spun down and suspended in PBS supplemented with CMP-NANA (50 µg/mL) to an OD_620 nm_ of 0.3. Bacteria were further diluted in cell culture media to reach ∼1 × 10^5^ CFU/mL. For the adherence assays, monolayers of human cervical (ME-180, ATCC #HTP-33) and vaginal epithelial (VK2/E6E7, ATCC #CRL-2616) cells were incubated at 37 °C, 5% CO_2_ for four hours with ~1 × 10^5^ CFU sialylated *N. gonorrhoeae* strains that had been pre-incubated with 25 µg/mL purified mouse IgG (prepared from 2 pools of serum from 4 mice) for 15 min in cell culture media. Purified IgG was used for in vitro colonization assays since an unknown factor, which was not IgM, in normal mouse serum and negative control serum from mice given adjuvant only inhibited adhesion of Ng strains to cells. Following infection, cells were washed three times with PBS to remove non-adherent bacteria; the cells were detached using Gibco™ StemPro™ Accutase™ (ThermoFisher Scientific, Waltham, MA, USA) and transferred to microcentrifuge tubes. Cells were washed twice with PBS and fixed with 4% Paraformaldehyde. Labeled adherent bacteria were quantitated using ImageStreamX (Cytek Biosciences, Fremont, CA, USA). Data are presented as percentage of inhibition of adherent bacteria relative to adjuvant-only antibody control.

Transgenic mice (*N* = 10 mice per group) were screened for the human CEACAM1 gene and human FH protein expression as described elsewhere [24]. MenB strain 8047 and Ng strain WHO F were grown as described for the SBA assay protocol above. After reaching an OD_620 nm_ of 0.6 for Nm and after being resuspended to an OD of 0.3 for Ng, bacteria were diluted to obtain to the targeted inoculum of 10^3^ CFUs/animal for 8047 and 10^7^ CFUs/animal for WHO F. Immediately after the challenge, a sample of the bacteria preparation was plated on an agar plate to confirm the inoculum. Mice were challenged intranasally and euthanized after 48 h for Nm challenge and 72 h for Ng challenge. After tracheal wash, swabs of hard palate and soft palate tissue samples were collected for each animal and plated on GC agar plates (ThermoFisher Scientific, Waltham, MA, USA, Cat# DF0289-17-3) supplemented with IsoVitalex (ThermoFisher Scientific, Waltham, MA, USA, Cat# B11876) and VCNT inhibitor (ThermoFisher Scientific, Waltham, MA, USA, Cat# B12408). Colonies were visible after 16–24 h incubation at 37 °C with 5% CO_2_. The tracheal wash was performed using an insulin syringe filled with 500 µL of PBS, with 1 mM Mg buffer inserted into the trachea, and the wash was collected from the nares into a 1.5 mL microtube. To obtain bacteria adhering to the nasal pharynx mucosal surfaces, the lower jaw of the mouse was removed. The soft palate was collected into a tube filled with 500 µL of 0.05% trypsin–EDTA or Accutase™ (ThermoFisher Scientific, Waltham, MA, USA). The hard palate and nasal passages were swabbed with an ultrafine aluminum shaft applicator with a calcium alginate fiber tip (Puritan, Guilford, ME, USA), which was resuspended into the tube with soft palate tissue sample. The soft palate and fiber tip were incubated at 37 °C for 10 min before plating the entire solution on a chocolate agar plate. The tracheal wash was plated on a separate plate. CFUs were counted the following day after overnight incubation at 37 °C in an atmosphere of 5% CO_2_. Statistical comparison between the results for mice given adjuvant only with mice immunized with the Nm-Ng NOMV vaccine was performed using a Welch’s two-tailed *T*-test with Prism (GraphPad 10.6.1).

## 3. Results

The IgG anti-FHbp titers of all CD1 WT mice before vaccination were below the lowest serum dilution tested (<1:100). At two weeks after dose 2, the IgG titers remained below 1:100 in the adjuvant negative control group and the Ng NOMV vaccine group. IgG titers increased in both vaccine groups (Figure 1a) where the Nm NOMV component was present (alone and in combination with the Ng NOMV vaccine). The anti-FHbp IgG titers elicited by the Nm-Ng NOMV combination were not significantly reduced compared to the Nm NOMV alone. To test the breadth of the SBA, MenB strains were chosen to represent both close (FHbp IDs 13 and 24) and distant (ID15 and ID28) homologies to the FHbp antigens present in the vaccines (Figure 1b) for FHbp Subfamilies B and A, respectively. The SBA titers using IgG/IgM-depleted human complement are shown in Figure 1c. There was no significant difference between titers elicited by the combination Nm-Ng NOMV vaccine or Nm NOMV alone. The Ng NOMV component and adjuvant-only antisera had no SBA against the four MenB test strains.

The IgG titers against the NOMV from two Ng strains, WHO M (PorB.1a) and WHO N (PorB.1b), that are heterologous to the vaccine NOMVs are shown in Figure 2a. The titers of mice given the Ng NOMV alone or in combination are approximately 10-fold higher than for mice given the Nm NOMV vaccine only, and there was no significant difference between the titers of mice immunized with the Ng NOMV or the Nm-Ng NOMV combination vaccine.

The Nm NOMV antisera had little or no SBA against the Ng test strains. Ng NOMV and Nm-Ng NOMV antisera had high SBA titers against strains expressing PorB.1a and PorB.1b. The difference in Ng NOMV SBA titers was not significantly different from those of the Nm-Ng NOMV vaccine except for the Ng strain FA1090, which is homologous to the strains used to prepare the Ng NOMV component. PorB.1b and PorB.1a in the strains used to prepare the NOMV vaccine were heterologous to PorB in WHO M, WHO G, and WHO N. The SBA data for MS11 is not included in Figure 2b because, even when sialylated, the strain was susceptible to bacteriolysis with the active complement alone or in combination with anti-alum control sera. The data is shown in Appendix A.

Currently, there is no in vitro correlate for protection against Ng infections. However, for N. gonorrhoeae to cause an infection, it must adhere to the mucosal epithelium. Therefore, we investigated whether purified IgG antibodies from mice immunized with the NOMV vaccines were able to inhibit the adherence to VK2/E6E7 vaginal and ME-180 cervical epithelial cells using sialylated, fluorescently labeled bacteria from Ng strains expressing PorB.1a (WHO F, WHO G, WHO M) and PorB.1b (FA1090, MS11, and WHO M). The data were obtained and analyzed using an ImageStreamX Mark II flow cytometer that combines flow cytometry with fluorescence microscopy to identify and count individual cells with adherent bacteria. The purified IgG from mice immunized with the Nm NOMV had significant inhibition activity against WHO G in VK2/E6E7 vaginal cells compared to the adjuvant-only control but not any other Ng strain or any strain in ME-180 cervical cells (Figure 3a,b). In contrast, the IgG from mice immunized with the Ng NOMV and the combination Nm-Ng NOMV vaccine inhibited the adherence of all Ng strains in both cell lines except for WHO M on VK2/E6E7 cells and FA1090 and WHO M on ME-180 cells (Figure 3a,b). The differences in the inhibitory activity between the purified IgG from the Ng NOMV- and Nm-Ng NOMV-immunized mice were not significant.

Ng is a human-restricted pathogen. Although a wild-type mouse Ng challenge model is commonly used to test whether vaccine-elicited antibodies can inhibit vaginal colonization by Ng strains, the wild-type mice used lack human-specific receptors and immune shielding mechanisms and, therefore, do not replicate the Ng colonization of human mucosal tissues. To determine whether mice immunized with NOMV vaccines can inhibit colonization in an animal model, we constructed a transgenic (Tg) mouse line that expresses human CEACAM1 for adhesion through Opa proteins and Factor H to provide immune shielding. The human CEACAM1/FH Tg mice were challenged intranasally with the MenB strain 8047 or the Ng strain WHO F after immunization with two doses of each NOMV vaccine or the adjuvant only. The Nm 8047 strain was heterologous to the strains used to prepare the NOMV used in the vaccines. For MenB colonization, an additional control included 4CMenB, which provides protection against MenB disease but does not appear to affect colonization in humans [25]. As shown in Figure 4, mice given the NOMV-NmNg vaccine demonstrated inhibited colonization of the MenB strain 8047, which was significant compared to the adjuvant-only control but not to 4CMenB. Also, Tg mice immunized with NOMV-NmNg demonstrated inhibited colonization of WHO F compared to mice immunized with the adjuvant or NOMV-Ng (*p* > 0.0001).

## 4. Discussion

To date, the only vaccines shown to have efficacy in preventing gonorrhea are vaccines containing vesicles isolated from the closely related bacterial species *Neisseria meningitidis*. Building upon this data, we constructed a novel vaccine, Nm-Ng NOMV, that combines a next-generation NOMV MenB vaccine with NOMVs produced from Ng strains that express PorB.1a and PorB.1b. We focused on PorB for both Nm and Ng strains since PorB is a major component of OMVs from both Nm and Ng. Further support for this approach comes from a recent study of human monoclonal antibodies produced from subjects vaccinated with 4CMenB that were reactive with Nm and Ng PorB epitopes as well as LOS [16]. The anti-PorB antibodies had SBA against Ng strains and provided cross-protection in a wild-type mouse vaginal challenge model of gonococcal infection. The PorB amino acid sequence from the production strain for the Nm NOMV is identical to PorB in the OMV component of 4CMenB. The two Ng strains used for the production of NOMVs were chosen based on PorB.1a and PorB.1b sequences where the surface-exposed variable loops were among the most common for Ng isolates of each PorB subfamily circulating world-wide [26].

OMVax’s Nm-Ng NOMV vaccine elicited antibodies that had SBA against diverse Ng strains. In our SBA assay, the bacteria were grown in the presence of CMP-NANA, which is scavenged from the human host and used by the bacteria to sialylate LOS. The sialylation of LOS results in bacteria that are considerably more resistant to SBA than strains grown without CMP-NANA [27]. SBA assays were performed using an IgG/IgM-depleted human complement, which more closely replicates SBA in humans compared to SBA achieved with the non-human complement [28]. The SBA confirms that the Nm-Ng NOMV vaccine elicits antibodies that recognize antigens on the bacterial surface of the strains tested that are present in amounts sufficient for robust SBA. SBA has not been established as a surrogate for protection against the disease but is likely to be important in preventing gonorrhea.

Nm and Ng are obligate human pathogens that use mechanisms for attachment (CEACAMs, CD46), invasion [29], and immune shielding (FH [30], TspB [31], Opa_CEA_ [32]) that specifically interact with human receptors. Antibodies elicited by vaccines (e.g., IgA and IgG) are present in secretions enveloping epithelial cells that are in direct contact with Nm and Ng during the earliest stages of infection and can prevent colonization and invasion [33]. Antibodies that interfere with the mechanisms of colonization protect the host during the initial stages of infection from more advanced stages of disease and potentially limit transmission between vaccinated and unvaccinated individuals (i.e., community immunity). The most cost-effective and widely used vaccines provide both individual and community protection.

Purified IgG from mice immunized with Ng NOMV or Nm-Ng NOMV were able to inhibit adherence in five out of six Ng strains to human vaginal and/or cervical cell lines. Interestingly, antibodies elicited by both vaccines inhibited the adherence of the Ng strain FA1090 to vaginal cells but not cervical cells and did not inhibit the adherence of the Ng strain WHO M to either cell line even though both vaccines produced high titers of bactericidal antibodies against WHO M (Figure 3a, b). Inhibiting adherence is also likely important for protection, but adherence mediated by the bacterial Opa protein binding to host CEACAM receptors is complicated by the expression of different CEACAMs (e.g., CEACAM1 in cervical tissues and CEACAM5 in vaginal tissues) and the expression of different Opa proteins by the bacteria, which also vary depending on environmental factors [34]. Although we did not quantitate antibodies to Opa proteins elicited by the Ng NOMV or Nm-Ng NOMV, it is possible that the presence of multiple Opa proteins in the mixture of NOMVs may be important for eliciting antibodies that can inhibit adherence of diverse strains in the vaginal and cervical cell culture models of Ng adherence.

To evaluate the ability to inhibit the Ng colonization in a live animal, we used a transgenic mouse model of Ng colonization that replicates some aspects of the pathogenesis in humans in expressing human CEACAM1, a receptor specific for Opa proteins expressed by Nm and Ng that mediates adhesion, and human FH to provide immune protection. The mice were challenged intranasally, which enabled us to test both male and female mice. Although the results were not statistically significant in this under-powered preliminary experiment, six of the seven mice vaccinated with the Nm-Ng NOMV and challenged intranasally with the Ng strain WHO F had no colonizing bacteria compared to only two of the seven mice given the adjuvant alone or one of the six given Ng NOMV alone (Figure 4). The trend suggests that the combination vaccine that included the MenB NOMV component provided superior protection compared to the Ng NOMV component alone, which is consistent with the results of the epidemiological studies showing vaccines with a MenB OMV provide some protection against gonorrhea in humans [1,3,4,5,6].

Although there are two MenB vaccines approved for use in humans in multiple countries, neither 4CMenB or MenB-FHbp is broadly protective or able to protect against the disease caused by an increasing number of strains currently causing outbreaks in several European countries and Canada [11,12,13]. The situation calls for the development of a next-generation MenB vaccine that can provide broader coverage. The Nm NOMV component of the Nm-Ng NOMV vaccine contains mutant FHbps from Subfamilies A and B with reduced FH binding overexpressed in NOMVs, such that the epitope structure and native environment is preserved. As a result, the FHbp antigens elicit high titers of antibodies that bind to FHbp and mediate complement-dependent bactericidal activity but can also block FH binding, which further enhances SBA [35]. The SBA is a surrogate for protection against meningococcal disease in humans [36]. As shown in Figure 1c, the ability of the Nm NOMV to elicit antibodies with broad SBA was not diminished by combining it with the Ng NOMV in the Nm-Ng NOMV vaccine. Similarly, the Nm-Ng NOMV elicited antibodies with high SBA titers against diverse Ng strains compared to the Ng NOMV alone (Figure 2) and inhibited the colonization of a heterologous MenB strain (8047) and a homologous Ng strain (WHO F) in a mouse model of meningococcal disease (Figure 4a).

Finally, a combination MenB-Ng vaccine may be advantageous with respect to acceptance and the potential to control the spread of gonorrhea. Adolescents and young adults are commonly vaccinated for protection against MenB, and this age group is also at the greatest risk of acquiring new gonorrhea infections [17]. The Nm-Ng NOMV vaccine has the potential to address both problems since a vaccine with the potential to prevent life-threatening meningococcal disease and gonorrhea in a population of sexually active subjects at highest risk of Ng infections may be more acceptable to the public than a vaccine specifically for the prevention of a sexually transmitted infection.

## Figures and Tables

**Figure 1 pathogens-14-00979-f001:**
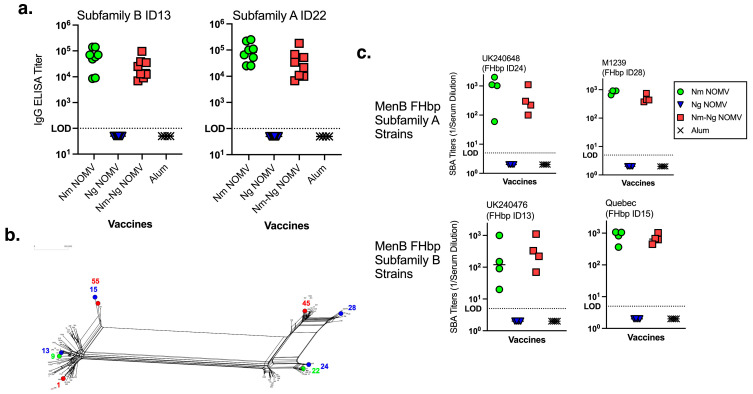
(**a**). Serum IgG anti-FHbp titers by ELISA (recombinant FHbp on the plate). Nm-Ng NOMV vaccine showed comparable IgG titers to the Nm NOMV-only vaccine. Each symbol represents a single animal. Bars represent the median. Nm NOMV and Nm-Ng NOMV IgG titers were not significantly different using an unpaired *T*-test analysis. (**b**). FHbp ID splits Tree dendrogram. FHbp sequence IDs for a set of currently circulating invasive MenB strains from the CDC are compared to those in Nm-Ng NOMV (shown in green), 4CMenB, and MenB-FHbp vaccines (shown in red). FHbp IDs for strains used for SBA are highlighted in blue. (**c**). SBA titers against MenB strains. SBA assays performed using IgG-IgM-depleted human serum as source of complement and bacteria grown in the presence of CMP-NANA. Nm-Ng NOMV vaccine showed similar SBA compared to Nm NOMV against MenB strains irrespective of matching or mismatching FHbp sequences. Serum from 4CMenB immunized mice had no SBA, except for Quebec and UK240476 strains (GMT 1:210 and 1:25, respectively). Each symbol represents a pool of two mouse sera. Bars represent the median. Nm NOMV and Nm-Ng NOMV SBA titers were not significantly different using an unpaired *T*-test analysis.

**Figure 2 pathogens-14-00979-f002:**
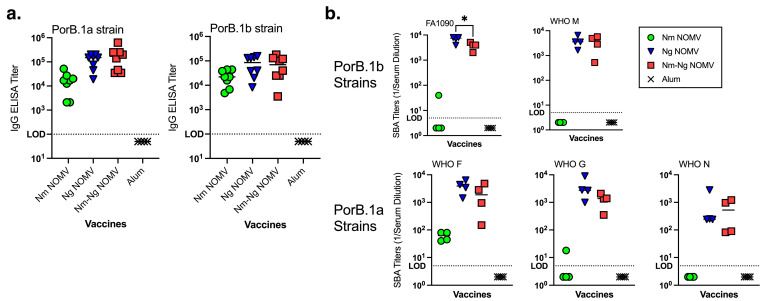
(**a**) Serum IgG anti-Ng titers by ELISA (NOMV from two heterologous Ng strains on the plate). Nm-Ng NOMV vaccine showed comparable IgG titers to the Ng NOMV-only vaccine. Each symbol represents a single animal. Bars represent the median. Ng NOMV and Nm-Ng NOMV IgG titers were not significantly different using an unpaired *T*-test analysis. (**b**) SBA titers against Ng strains. SBA assays performed using IgG-IgM-depleted human serum as source of complement and bacteria grown in the presence of CMP-NANA. Nm-Ng NOMV vaccine elicited high SBA titers, similar to the Ng NOMV vaccine, against a broad panel of Ng strains. Serum from 4CMenB-immunized mice had no SBA against FA1090, and it was not tested against other Ng strains. Each symbol represents a pool from two mouse sera. Bars represent the median. Significant P-values from unpaired, two tailed T-tests are indicated by * ≥ 0.5. Ng NOMV and Nm-Ng NOMV SBA titers were not significantly different.

**Figure 3 pathogens-14-00979-f003:**
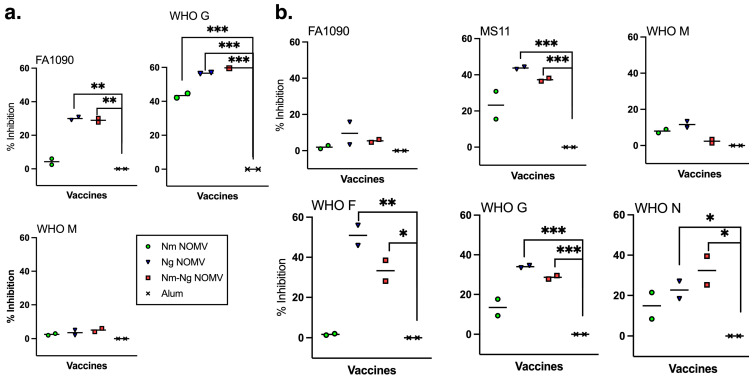
Inhibition of adhesion by Ng strains to cervical and vaginal human cells by IgG purified from serum pools. Sialylated bacteria and 25 µg/mL of purified IgG were added to VK2/E6E7 or ME-180 cells to test antibody’s ability to prevent Ng adhesion to human cells. Adhering labeled bacteria were quantified using ImageStreamX. Test strains included both PorB.1a (WHO F, WHO G, and WHO N) and PorB.1b strains (FA1090, MS11, WHO M). (**a**) Inhibition of adherence to VK2/E6E7 vaginal cells (MS11, WHO M, and WHO F did not adhere to VK2/E6E7 cells) and (**b**) ME-180 cervical cells. Each symbol represents a pool from four mice’s purified IgG. Bars represent the median. Significant P-values from unpaired, two tailed T-tests are indicated by * ≥ 0.5, ** ≥ 0.01, and *** ≥ 0.001.

**Figure 4 pathogens-14-00979-f004:**
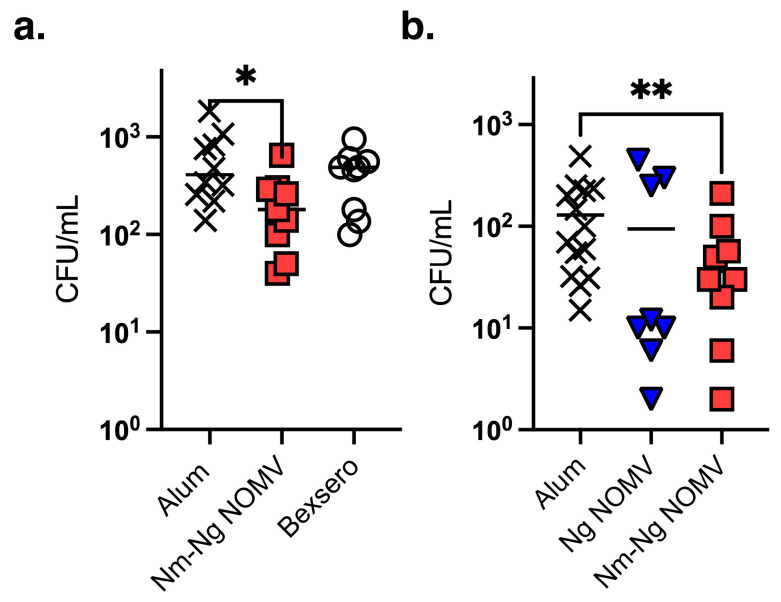
In vivo inhibition of colonization. (**a**) MenB colonization. Transgenic (Tg) mice (*N* = 8) immunized with either Nm-Ng NOMV, Bexsero, or alum were challenged with 1 × 10^3^ CFUs of heterologous MenB strain 8047. Bacteria were recovered after 48 h from the challenge. Each symbol represents a single animal. Bars represent the median. Nm-Ng NOMV decreased CFUs in the nasopharynx compared to the alum control (*p* = 0.04). (**b**) Ng colonization. Tg mice (*N* = 10) immunized with either Ng NOMV, Nm-Ng NOMV, or alum were challenged intranasally with 1 × 10^7^ CFUs of Ng strain WHO F. Bacteria were recovered after 72 h from the challenge. The difference between the alum control and the Tg mice receiving the NOMV vaccines was statistically significant (*p* = 0.0055) using a Mann–Whitney test. * ≥ 0.5, ** ≥ 0.01.

## Data Availability

All the data described is included in this manuscript.

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
