# Peer review of "A Combination Native Outer Membrane Vesicle (NOVM) Vaccine to Prevent Meningococcal and Gonococcal Disease"

_pathogens, 2025, doi:10.3390/pathogens14100979_

Round 1
Reviewer 1 Report
Comments and Suggestions for Authors
Comments to Author
Title: A COMBINATION NATIVE OUTER MEMBRANE VESICLE (NOMV) VACCINE TO PREVENT MENINGOCOCCAL AND GONOCOCCAL DISEASE
Authors: Serena Giuntini, Scarlet W. Tefera, Alejandro Bolanos, Adan Ramos Rivera, Gregory R. Moe
Overview and general recommendation
This manuscript manuscript presents a compelling development of a combined vaccine based on NOMV derived from genetically modified strains of Neisseria meningitidis and Neisseria gonorrhoeae. The data provided convincingly demonstrate that antibodies elicited by the combined Nm-Ng NOMV vaccine not only exert bactericidal activity against various Nm and Ng strains cultured with CMP-NANA but also inhibit Ng adhesion to human cervical and vaginal cells, and reduce colonization by both pathogens in a transgenic mouse model. These findings underscore the vaccine's potential to protect against disease and limit colonization by both Nm and Ng, representing a significant stride toward an effective dual-target vaccine.
The manuscript is well written, and the experiments/investigations are explained clearly and easy to follow.
This manuscript may be accepted once the following points are addressed:
- line 48, Please correct “Phase 2” to “Phase 1/2”
- line 49-50, Since this information is unsubstantiated, please remove the final part “and it is unclear whether the vaccine will continue to be developed by the company”
- Materials and Methods:
- No Ng strains are reported in the study. However, line 242 of the results section mentions FA1090. Were the NOMV Ng derived solely from the FA1090 strain, or were additional strains involved?
- Results – Adhesion Inhibition Assay:
- Why do the authors use purified IgG in the inhibition of adhesion (were whole sera inadequate?) while using sera in the SBA? Please clarify the rationale behind using purified IgG in one assay and whole sera in the other.
- Additionally, regarding Alum as a control, is the data based on purified IgG in the adhesion inhibition assay, or is it derived from serum?
- Could the authors please clarify the rationale behind some of the results? It seems that the outcomes are either negative or positive depending on the strain or cell line, but a corresponding scientific explanation is not evident.
- Results – in vivo inhibition of colonization assay:
- In the animal colonization model, the authors reported results for Bexsero only regarding the colonization by the meningococcal strain. Was a similar analysis performed for Neisseria gonorrhoeae? If not, could the authors explain why this analysis was not included?
These revisions should help clarify the issues and improve the overall presentation of your manuscript.

Reviewer 2 Report
Comments and Suggestions for Authors
The authors developed a combination MenB NOMV vaccine component (Nm NOMV) with overexpressed mutant FHbp from subfamilies A and B. They have also produced combination vaccine referred to as Nm-Ng NOMV and tested them in mice. They have demonstrated Nm-Ng NOMV can induce SBA and adhesion-blocking IgG and reduce the colonization of MenB. As there is no licensed Ng vaccine yet, this paper will contribute a lot to the progress of this field. Specific comments follow.
Major points:
- Line 159: Please explain why the authors used only anti-mouse κ chain specific antibody but not anti-mouse γ chin antibody to measure IgG concentrations.
- Line 188: Please indicate the source catalogue numbers of the reagents so that the readers can follow your methods.
- Please include ethics statement.
Minor points:
- Lines 113, 118: Please use consistent description of “dose (Dose)”.
Reviewer 3 Report
Comments and Suggestions for Authors
Humans are the sole host for Neisseria gonorrhoeae, and appropriate use of vaccines could help eradicate the pathogen. However, no vaccine has been successfully developed that effectively prevents or treats gonorrhea infection today. Sexually transmitted diseases are proving increasingly difficult to treat as the bacteria involved develop greater resistance to standard antibiotics. Even after successful treatment, individuals may become reinfected. Antibiotic-resistant gonorrhea is emerging as a growing concern.
Research indicates that certain antibodies produced by meningitis vaccines can bind to gonorrhea bacteria. The current meningococcal vaccine 4CMenB was introduced in Australia in 2019 for individuals aged 17 to 20. Marshall's research team compared gonorrhea infection rates between vaccinated and unvaccinated individuals. Results suggest the vaccine's effect is relatively modest, reducing sexually transmitted infection rates by only 40%.
The study developed a combined vaccine based on natural outer membrane vesicles (NOMV) from two strains of Neisseria meningitidis (Nm) and two strains of Neisseria gonorrhoeae (Ng). Compared to existing group B meningococcal vaccines, it provides broader coverage and higher serum bactericidal activity. It induces broad-spectrum SBA antibodies against group B and gonococcal strains, inhibits gonococcal colonization in human cervical and vaginal cell lines, and demonstrates inhibitory effects in the gonococcal transgenic mouse model.
The paper demonstrates rigorous logic and presents clear conclusions. It provides experimental data supporting vaccine research in the future that simultaneously prevents meningococcal disease and gonorrhea in both adolescents and adults.
The following discussion addresses only certain specific points and does not affect the overall evaluation of the paper.
1. Figure 2b lacks data for the MS11 strain, and Figure 3a lacks data for the MS11, WHO F, and WHO N strains. It would be best to complete these data for comparative analysis.
2. Due to the absence of the aforementioned data or figures, the comparison of inhibitory effects across different strain in colonization experiments remains incomplete and confusing. Although the paper discusses the results for strain FA1090 and the ineffective results for the WHO M strain in lines 335–347, Figure 3 currently shows that only the results for the WHO G strain among the six strains align with the paper's expectations.
3. In the Figure 2 correspondingly, during the serum bactericidal activity (SBA) assay, do the Ng NOMV and Nm-Ng NOMV antisera also exhibit high-titer SBA against the MS11 strain?
Reviewer 4 Report
Comments and Suggestions for Authors
I was invited to revise the paper entitled "A COMBINATION NATIVE OUTER MEMBRANE VESICLE (NOMV) VACCINE TO PREVENT MENINGOCOCCAL AND GONOCOCCAL DISEASE". This study aimed to demonstrate that a combination MenB NOMV vaccine component with overexpressed mutant FHbp from subfamilies A and B provides higher bactericidal activity against MenB.
THe topic is interesting and this work is relevant for public health.
Methods are clearly presented.
Main criticism:
- No statistical analysis was presented and it limits the understanding of results. So, this point needs to be addressed in order to better evaluate the paper;
- English language needs a deep revision;
- Figures need to be larger and its quality should be improved;
- The evaluation of the activity against gonococcal dieases was marginal. Also MenB vaccination, despite its collateral activity, han no reccomendations against it;
- Sample size estimation was lacking
-
Round 2
Reviewer 3 Report
Comments and Suggestions for Authors
I appreciate the authors' thorough response to each review comment. Their prompt completion of supplementary figure demonstrates their command of the original data, clear experimental design, and understanding of the project's implementation status.
The combined vaccine developed in this study provides experimental data supporting its potential applications in preventing both meningococcal disease and gonorrhea in adolescents and adults.
Reviewer 4 Report
Comments and Suggestions for Authors
It can be accepted.